# Climatology and Trends in Concurrent Temperature Extremes in the Global Extratropics

Gabriele Messori[1,2,3], Antonio Segalini[1], Alexandre M. Ramos[4]

[1] Department of Earth Sciences, Uppsala University, Uppsala, Sweden.
[2] Swedish Centre for Impacts of Climate Extremes (climes), Uppsala University, Uppsala, Sweden.
[3] Department of Meteorology and Bolin Centre for Climate Research, Stockholm University, Stockholm, Sweden.
[4] Institute of Meteorology and Climate Research, Karlsruhe Institute of Technology, Karlsruhe, Germany.

*Correspondence to:* Gabriele Messori (gabriele.messori@geo.uu.se)

**Abstract.** Simultaneous occurrences of multiple heatwaves or cold spells in remote geographical regions have drawn considerable attention in the literature, due to their potentially far-reaching impacts. We introduce a flexible toolbox to study such concurrent temperature extremes, with adjustable parameters that different users can tailor to their specific needs. We then use the toolbox to present a climatological analysis of spatially compounding heatwaves and cold spells in the global midlatitudes. Specific geographical areas, such as Western Russia, Central Europe, Southwestern Eurasia and Western North America, emerge as hotspots for concurrent temperature extremes. Concurrent heatwaves are becoming more frequent, longer-lasting and more extended in the Northern Hemisphere, while the opposite holds for concurrent cold spells. Concurrent heatwaves in the Southern Hemisphere are comparatively rare, but have been increasing in both number and extent. Notably, several of these trends in concurrent temperature extremes are significantly stronger than the corresponding trends in all temperature extremes.

## 1. Introduction

Extreme climate events often do not occur in isolation, but are triggered by complex processes leading to multiple extremes. An example are events occurring roughly simultaneously at remote locations. These are typically due to specific large-scale atmospheric or oceanic features and are referred to as concurrent, or spatially compounding, extremes (Zscheischler et al., 2020). Temperature extremes often play a prominent role in concurrent events, either in isolation or in conjunction with other extreme event categories. Specific examples in the Northern Hemisphere (NH) include drought–cold spells–wet and windy extremes during winter 2013/14 (Davies, 2015), heatwave–heavy precipitation during summer 2010 (Lau and Kim 2012; di Capua et al., 2021) and concurrent heatwaves and heatwave–heavy precipitation during summer 2018 (Kornhuber et al., 2019). Building upon these episodic events, the literature has considered recurrent spatially compounding extremes, such as wintertime cold–wet–windy extremes in North America and Europe (Messori et al., 2016; Leeding et al., 2023; Riboldi et al., 2023; Messori and Faranda, 2023), and opposite temperature extremes in East Asia and North America (Sung et al., 2021). A parallel line of work has analysed large-scale atmospheric patterns favouring concurrent extremes, highlighting the role of large-amplitude or recurrent atmospheric waves (Coumou et al., 2014; Röthlisberger et al., 2019; Kornhuber et al., 2020; Bui et al., 2022; White et al., 2022; Kornhuber and Messori, 2023). These waves are particularly effective in engendering temperature extremes in both the warm and cold seasons (e.g. Screen and Simmonds, 2014), whose synchronised occurrence triggers detrimental socio-economic and environmental

impacts such as widespread crop failures (Tigchelaar et al., 2018; Kornhuber et al., 2020; Gaupp et al., 2020), increased mortality, wildfires, power supply disruptions and more (Vogel et al., 2019).

Previous work on concurrent temperature extremes has often focussed on specific regions or seasons (e.g. Röthlisberger et al., 2019; Kornhuber et al., 2019). More recent work has provided an overview of long-term trends and geographical hotspots of concurrent extremes (Rogers et al., 2022), yet with a focus on NH heatwaves

and for a fixed definition of what is "extreme". Broader studies looking at compound extremes or multi-risks have not tailored their analyses and methods specifically to temperature extremes, providing limited flexibility for defining both the extremes themselves and their spatial relation (e.g. Claassen et al., 2023). In this study, we pursue a dual aim. We first introduce a flexible toolbox to compute statistics on concurrent temperature extremes, with adjustable parameters that different users can tailor to their specific needs. We then use the toolbox to present

a climatological and trend analysis of concurrent hot and cold extremes in both hemispheres. While the toolbox was developed specifically for temperature extremes, it can in principle take any single-level variable as input. We provide a proof-of-concept for application to 10m wind extremes in Appendix B.

## 2.   A flexible toolbox for the analysis of concurrent temperature extremes

*Input and parameters* – Our toolbox takes as input gridded temperature (or temperature anomaly) data on a regular latitude-longitude grid. The users first need to: (i) define a latitudinal domain; (ii) pick a season in the form of a set of months; (iii) choose whether to limit the analysis to land gridpoints or not; and (iv) select a percentile to define the temperature extremes. After this, the toolbox computes the percentile threshold at each gridbox and identifies gridpoints above (for heatwaves) or below (for cold spells) said threshold (Fig. 1a). Next, the users can

impose a minimum duration requirement, such that only gridpoints exceeding the threshold over a set number of consecutive days are retained (Fig. 1b).

*Clustering and minimum extent* – In the following step, the users can flexibly define how to cluster different connected areas of extremely hot or cold gridpoints. Often, one finds a large connected area of extremely hot or cold temperatures, with smaller surrounding areas separated by a few gridpoints from the main area. These are

likely caused by the same physical driver(s) as the main connected area. For example, there is no *a priori* reason to believe that two regions of temperature extremes with a one-gridpoint gap in-between should be associated with independent large-scale drivers. The users can decide not to cluster different connected areas, to cluster them based on the distance between the locations of each connected area's centroid, or to cluster them based on the distance between the closest gridboxes of each pair of connected areas. The output of the clustering is a set of

well-separated temperature extreme regions (Fig. 1c). We are aware of existing clustering algorithms, which have been used in the context of compound climate extremes (e.g. Tilloy et al., 2022). We do not claim that our algorithm outperforms these, but rather that it provides a more intuitive parameter input set, in terms of type of distance in kilometres (centroids or closest points), as opposed to quantities such as minimum density of points in a given neighbourhood. Finally, the user can impose a minimum areal extent for each clustered temperature

extreme region (Fig. 1c). Given that our toolbox does not consider set regions, but rather updates the boundaries of the temperature extremes at every timestep, we do not consider time lags in the compounding, but rather output the statistics at every timestep of our data (see also Rogers et al., 2022, for a similar argument).

We recognise that our toolbox comes with a number of conceptual simplifications, for example by ignoring the shape of the connected extreme temperature areas when performing the clustering (beyond implicitly taking it

into account if computing centroids) and by imposing persistence as continuous exceedance of a threshold instead of allowing for discontinuous exceedances clustered in time. These were decided upon to strike a balance between flexibility and interpretability/usability of the toolbox.

## a) Define domain, season, threshold and other parameters

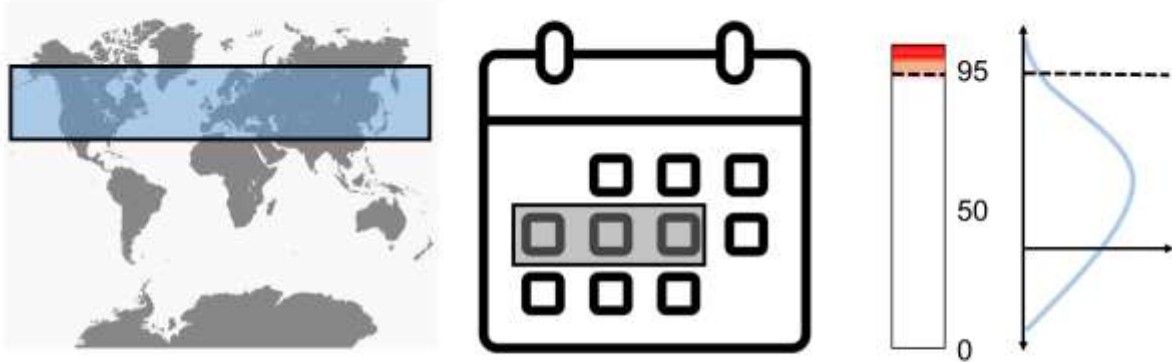

### b) Determine minimum duration          c) Cluster and impose minimum extent

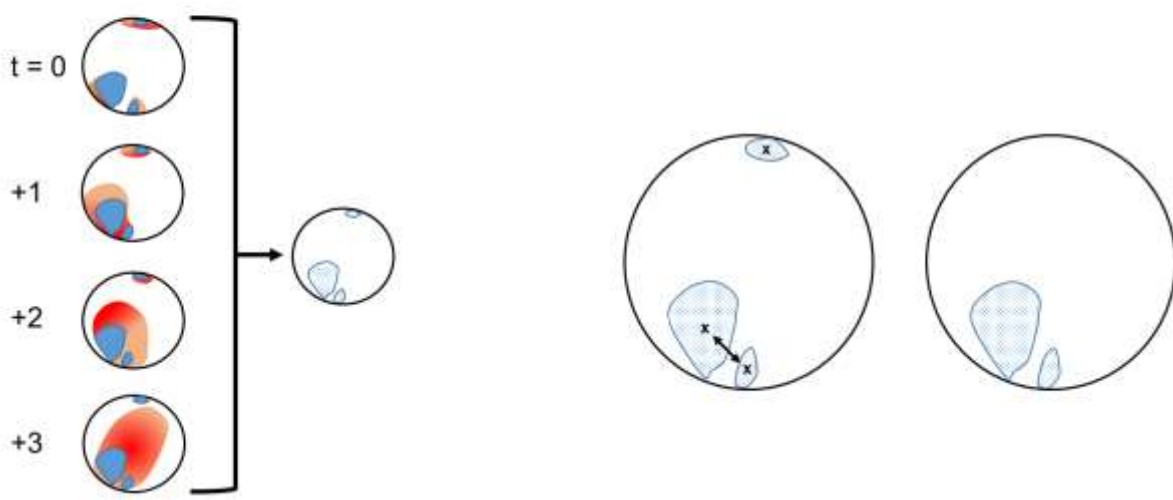

**Figure 1:** Schematic of the concurrent temperature extremes algorithm. (a) Given a latitudinal domain, a season
and a temperature percentile (in the example here, the 95th percentile), the algorithm identifies temperature extremes. (b) A minimum duration in days is then imposed at every location (in the example here, 4 days). The red areas represent percentile exceedances. The overlaid blue shading, regions where the exceedances satisfy the minimum duration. (c) Connected extreme temperature areas are identified (stippled regions), and clustered (in this example using centroid distances, with centroids of each connected area marked by "x"). The two centroids
that are clustered are connected by a double-headed arrow. After clustering, a minimum area threshold is applied. The map in (a) is reproduced from FreeVectorFlags.com under an attribution-only licence. The calendar icon in (a) is reproduced from DinosoftLabs under an attribution-only licence.

In the analysis we present in this study, we use daily-mean ERA5 (Hersbach et al., 2020) 2-metre temperature data over 30°–70° N/S in the period January 1940–August 2023, with a horizontal resolution of 0.5° latitude and
95 longitude. However, the toolbox is easily applicable to other gridded climate datasets with different resolutions. We consider non-detrended temperature anomaly data, with anomalies computed relative to a daily climatology smoothed with a 15-day running mean. The toolbox however provides the option to perform a detrending step. We identify hot extremes during summer (boreal: June, July and August – JJA; austral: December, January and February – DJF) and cold extremes during winter (boreal: DJF; Austral: JJA). In Sect. 3, we consider percentile

thresholds of 95 (5) for hot (cold) extremes (similar to e.g. Harnik et al., 2016 and Guirguis et al., 2018). We also display results for percentile thresholds of 90 (10) in Appendix A, as these have also often been adopted in the literature (e.g. Peings et al., 2013; Lin et al., 2022 and Holmberg et al., 2023). The percentiles are applied to the temperature anomalies at each gridbox. We impose a minimum duration threshold of 4 days, again in line with previous work that often adopts thresholds of 3–5 days (e.g. Xu et al., 2016; Brown, 2022; Lin et al., 2022), and show here the results for clustering based on centroid distances below 1000 km. We further enforce a minimum areal extent of clustered temperature extremes of $2\times10^5$ km$^2$. This is an intermediate value compared to previous literature, which has considered thresholds from order $10^5$ to order $10^6$ km$^2$ (cf. Lyon et al., 2019 and Rogers et al., 2022). The parameters used in Sect. 3, chosen to enable a comparison of our results to previous literature on temperature extremes, are summarised in Table 1. The table also summarises the parameter sweep of the toolbox as shown in Appendix C.

**Table 1:** Toolbox parameters used for the analysis conducted in Sect. 3 and ranges tested in the parameter sweep shown in Appendix C.

| Parameter name | Parameter value in Sect. 3 | Range tested in Appendix C |
|---|---|---|
| Latitudinal domain | 30°–70° N/S | – |
| Season | DJF, JJA | – |
| Percentile for temperature extremes | 5th (cold spells) / 95th (heatwaves) | 1st – 15th / 85th – 99th |
| Minimum duration | 4 days | 2 – 9 days |
| Minimum cluster separation | 1000 km | 500 – 4000 km |
| Minimum areal extent | $2\times10^5$ km$^2$ | $1\times10^5$ km$^2$ – $5\times10^5$ km$^2$ |

### 3. A Global Climatology of Concurrent Temperature Extremes

Two clear hotspots for the occurrence of concurrent heatwaves emerge in Western Russia and Western North America (Fig. 2a), with relative frequencies around or exceeding 0.4. In other words, roughly 40% of single-gridbox heatwaves at those locations are part of a set of multiple concurrent, large-scale heatwaves in the NH. Heatwaves in Central Asia and Central North America instead typically occur in isolation. Concurrent cold spells display a different geographical distribution, with Northwestern North America, Central Europe and Southwestern Eurasia showing the highest relative occurrences (Fig. 2b). At some of these locations, roughly half of the single-gridbox cold spells are part of a set of multiple concurrent, large-scale events. Concurrent temperature extremes are rare in relative terms in the Southern Hemisphere (SH), presumably because of the much smaller landmass extent.

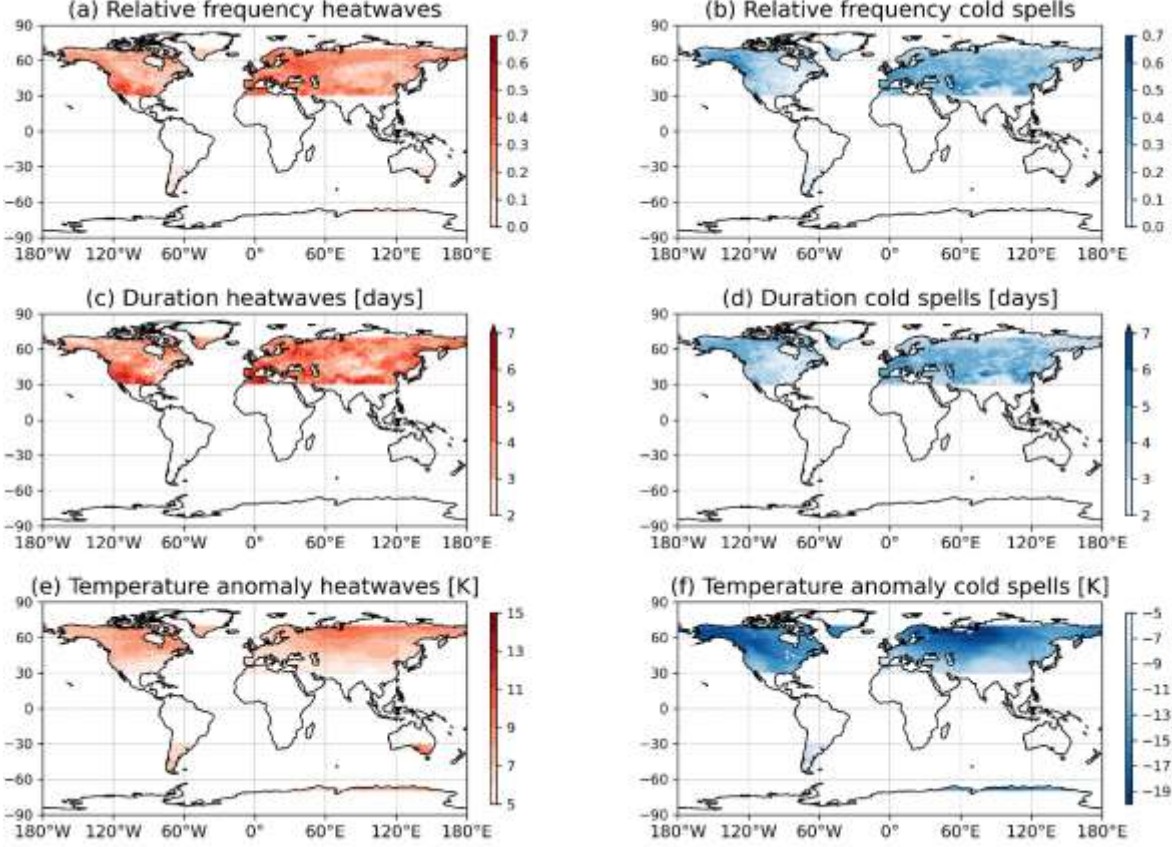

**Figure 2:** Relative frequency, duration and severity of concurrent temperature extremes. Relative frequency of concurrent (a) heatwaves and (b) cold spells, normalised relative to the percentile used to define the extremes. For example, a value of 0.5 means that half of the temperature extremes at a given location concur with another remote temperature extreme of the same sign. Mean duration (days) of concurrent (c) heatwaves and (d) cold spells. Mean temperature anomalies of concurrent (e) heatwaves and (f) cold spells. Panels (e) and (f) have differing colour range amplitudes.

The duration of both concurrent heatwaves and cold spells ranges in most regions between 3 and 6 days. The above-mentioned regional hotspots for occurrence frequency also display slightly above-average duration values. The rare concurrent temperature extremes in the SH show a much shorter duration than their NH counterparts. Note that, while we impose a minimum duration threshold for single-gridbox extremes, the duration of a compound temperature extreme can be shorter than this. Indeed, for compound extremes we only count the duration while there are two or more concurrent extremes – even if this temporal overlap period is shorter than the minimum duration threshold for single-gridbox extremes. The pattern of climatological temperature anomalies during the extremes roughly matches that of the temperature variance (Fig. A1), being largest in the Northern high latitudes and central parts of the continents, and larger for wintertime cold spells than summertime heatwaves. The corresponding results for a less stringent percentile definition of temperature extremes are shown in Fig. A2 and commented on in Appendix A.

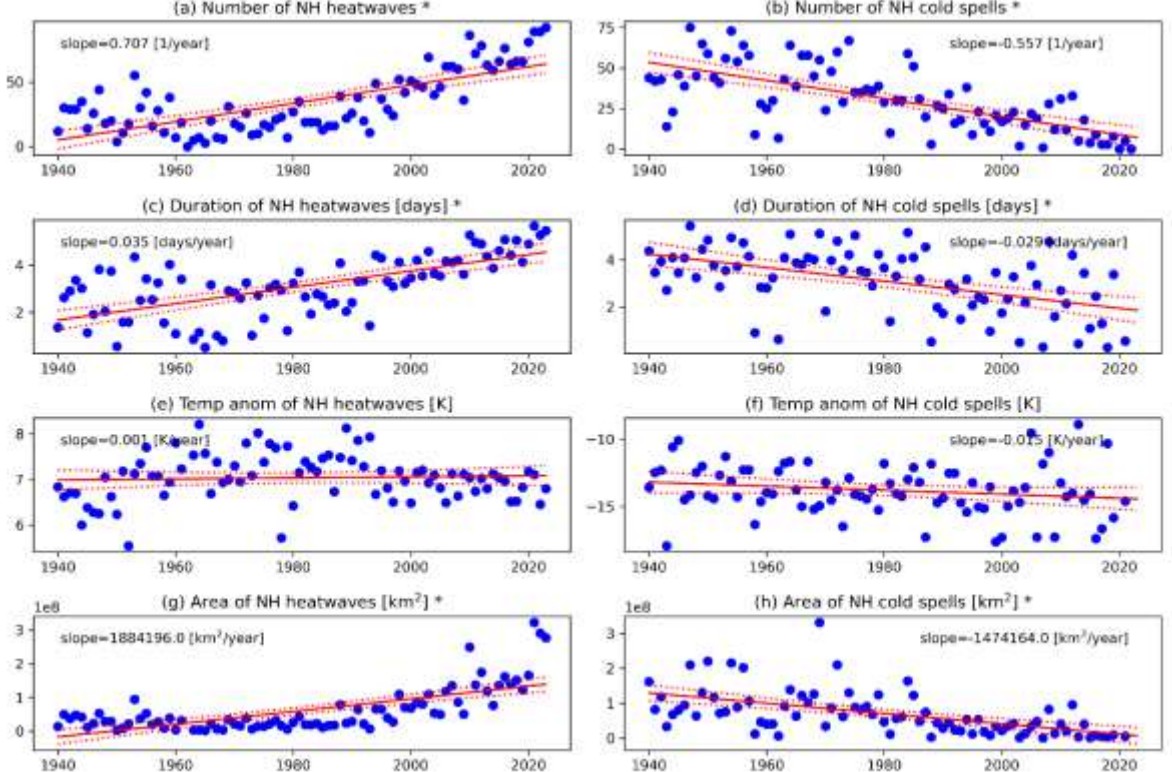

**Figure 3:** NH aggregated yearly data and trends in the occurrence, duration, severity and extent of concurrent temperature extremes. Number of concurrent (a) heatwaves and (b) cold spells. Mean duration (days) of concurrent (c) heatwaves and (d) cold spells. Mean temperature anomalies (K) of concurrent (e) heatwaves and (f) cold spells. Cumulative area ($km^2$) of concurrent (g) heatwaves and (h) cold spells. The continuous lines show linear fits, and the dashed lines 95% confidence bounds. The numbers in each panel show the linear fit slope. Asterisks in the panel titles indicate that the slope is different from 0 at the 5% level according to the *p*-value of the *t*-statistic. Seasons without any events are not accounted for when computing the linear fits for duration, severity and extent.

This climatology can be compared to that of all temperature extremes, regardless of whether they concur with others or not, but subject to the same percentile, duration and extent thresholds (Fig. A3). The spatial distributions are remarkably similar, with the main difference being that SH temperature extremes emerge more clearly. Moreover, the extremes display a longer duration since this now also includes days when they occur in isolation. We next consider trends in the concurrent temperature extremes, aggregated at hemispheric level (Fig. 3). Concurrent NH heatwaves have significantly increased in number, duration and area over the last 8 decades (Fig. 3a, c, g). As expected given our percentile-based definition, the temperature anomalies associated with the heatwaves have stayed roughly constant (Fig. 3e). Concurrent cold spells mostly mirror these trends, with significant decreases in number, duration and area (Fig. 3b, d, h), and no significant trend in temperatures. However, there are regions where both sets of temperature extremes show similar trends, for example in parts of Central Asia and Siberia, central North America and Scandinavia. The corresponding figures for the SH and for a less stringent percentile definition of temperature extremes are shown in Figs. A4–A6 and commented on in Appendix A.

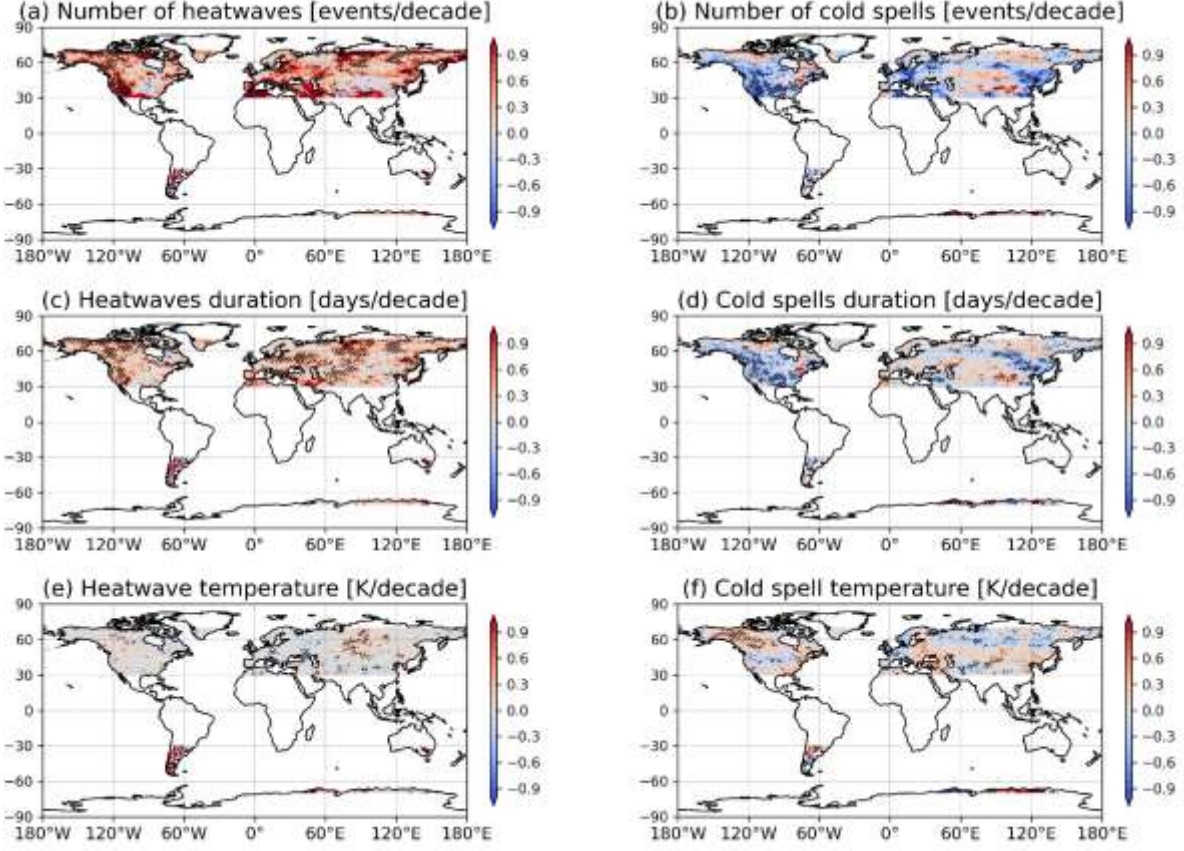

**Figure 4:** Linear trends in the occurrence, duration and severity of concurrent temperature extremes. Trends in number of concurrent (a) heatwaves and (b) cold spells; in duration of concurrent (c) heatwaves and (d) cold spells; and in temperature anomalies during concurrent (e) heatwaves) and (f) cold spells. Stippling shows regions where the trends are different from 0 at the 5% level according to the *p*-value of the *t*-statistic. A Gaussian filter with filter size 1 has been applied to all panels to improve legibility.

The above trends at hemispheric level can be compared to the corresponding trends for temperature extremes with no requirements for concurrence (Figs. A7, A8). The trends are qualitatively similar, with a marked increase in heatwaves and decrease in cold spells. We additionally see a significant decrease (increase) in NH (SH) cold spell temperature anomalies.

We conclude our analysis by considering trends on a geographical basis (Fig. 4). The frequency of concurrent heatwaves (cold spells) shows a widespread increase (decrease) across the global midlatitudes (Fig. 4a, b). The hotspot region of Western North America, already highlighted in Fig. 2, emerges as having a particularly strong and geographically widespread increasing heatwave trend. A strong decreasing trend is found for cold spells over the same region. Central Asia and Central North America are exceptions, showing little or even weak negative (positive) changes in the number of concurrent heatwaves (cold spells). The duration trends of concurrent temperature extremes roughly match the occurrence trends, albeit with some regional differences (Fig. 4c, d). For example, Western Russia emerges as having a significant positive duration trend in concurrent heatwaves but no significant occurrence trend. Finally, only a weak signal is seen in the temperature anomaly trends, with the most notable features being positive trends in South America for concurrent heatwaves and positive trends in Northwestern North America for concurrent cold spells. The corresponding figures for a less stringent percentile

definition of temperature extremes and for all extremes are shown in Figs. A9 and A10 and commented on in Appendix A.

### 4. Concluding Remarks

We have presented a flexible toolbox for the study of concurrent temperature extremes, and have applied it to cold and hot extremes in the global extratropics. The toolbox supports both climatological and trend analyses, and enables users to tailor the definition of concurrent temperature extremes to their specific needs. Indeed, different users may be interested in concurrent temperature extremes on a variety of different spatial and temporal scales – think for example of studies looking at heatwave impacts on the global food supply versus studies looking at regional or continental-scale health impacts of temperature extremes. While the toolbox was developed specifically for temperature extremes, it can in principle take any single-level variable as input. It is however not tailored to the detection of the dynamical drivers of the extremes.

Concurrent temperature extremes affect all land areas, although clear hotspots emerge, for example in Western Russia and Western North America for heatwaves and in Northwestern North America, Central Europe and Southwestern Eurasia for cold spells.

Consistent with previous work, we find clear upward trends in concurrent heatwave extent and number (Rogers et al., 2022), while decreasing trends are found for concurrent cold spells. For the concurrent NH heatwaves, the upward trend in occurrence corresponds to more than a quadrupling of events over 8 decades, while concurrent NH cold spells have become rarer by roughly two-thirds. Concurrent temperature extremes have historically mainly occurred in the NH extratropics, due to the much smaller land surface area in the SH. However, the sharp increase in concurrent heatwaves means that these are becoming an emerging hazard in the SH. We further find an increase in duration of concurrent heatwaves and a decrease for concurrent cold spells. In the first decades of the analysis period, concurrent cold spells lasted on average longer than concurrent heatwaves, while the opposite becomes true in the latter part of the analysis period. This is consistent with Lhotka and Kyselý (2015), who found a longer average duration for individual historical cold spells than heatwaves over Europe, and with Allen and Sheridan (2016), who found that in recent decades hot temperatures have on average become longer-lasting than cold temperatures in major U.S. cities. These multi-decadal trends are superimposed on interannual to interdecadal variability, consistent with the know link of several large-scale modes of climate variability with temperature extremes (e.g. Della Marta et al., 2017; Arblaster and Alexander, 2012; Loikith and Broccoli, 2014; Grotjahn et al., 2016, and references therein). On a geographical basis, we find regionally contrasting trends in the number, duration and temperature anomalies of concurrent cold spells in North America and Eurasia. These may be related to the Warm Arctic/Cold Continents pattern (Chen et al., 2018), which also modulates the occurrence of cold extremes (Ye and Messori, 2020). We further find that weaker cold spells show a more uniform decrease than more extreme cold spells (see Appendix A), suggesting that these regional patterns may be sensitive to the definition of extreme events.

Nonetheless, caution should be exercised in relating our results on concurrent extremes to those in the literature for extremes without any requirement for concurrence, as the trends can differ markedly. Indeed, the number, duration and extent of NH concurrent heatwaves have increased significantly more than the corresponding quantities for all NH heatwaves, and similarly for the decreases in cold spells (Fig. A11).

Given the flexibility of the algorithm presented here, it is our hope that it may be applied to a wide range of datasets, including climate projections, to better quantify the recent and future trends in concurrent temperature extremes and identify specific risk areas. It would be of particular interest to investigate whether the accelerated trends in concurrent temperature extremes relative to all temperature extremes may be associated with dynamical

trends associated with climate change. Indeed, a number of studies have found a connection between changes in the atmospheric circulation and regional heatwaves, concurrent heatwaves and long-term temperature trends (Cahynová and Huth, 2016; Rogers et al., 2022; Faranda et al., 2023; Vautard et al., 2023).

Concurrent heatwaves have thus become an increasingly frequent and widespread global hazard. Their increase is faster than that of all heatwaves. Concurrent cold spells show a rapidly decreasing trend, but still occur in the

present-day climate.

**Appendix A**

We provide below figures corresponding to those shown in the main manuscript, but for a different threshold definition of temperature extremes (Figs. A2, A5, A6, A9); temperature extremes with no requirements for

concurrence (Figs. A3, A7, A8, A10); and for the SH (Figs. A4, A6, A8). We further present a map of temperature variance in both the winter and summer seasons in each hemisphere (Fig. A1) and a figure comparing trends in concurrent and all temperature extremes (Fig. A11).

The qualitative features of Fig. 2 as discussed in the main text are reproduced in Fig. A2, although the concurrent temperature extremes display higher relative frequencies, are longer-lasting, and display weaker temperature

anomalies. These are intuitive results of including more, weaker events in our analysis.

Due to the small sample size, the only significant trends for the SH concurrent extremes (Fig. A4) are the increases in the number and area of concurrent heatwaves. The results for a less stringent percentile (Figs. A5, A6) reflect those shown in Figs. 3 and A4, albeit with higher numbers and areas of concurrent temperature extremes and weaker average temperature anomalies. Moreover, likely thanks to the larger sample size, we find significant

negative trends in the number, duration and extent of SH cold spells.

The key qualitative features of Fig. 4 as discussed in the main text are reproduced in Fig. A9, although in the latter figure the occurrence trends are generally stronger. Moreover, the negative trends in number and duration of cold spells are more geographically uniform. The trends for extremes without any requirement for concurrence (Fig. A10) are generally weaker than those for the concurrent extremes, notably for occurrence and duration.

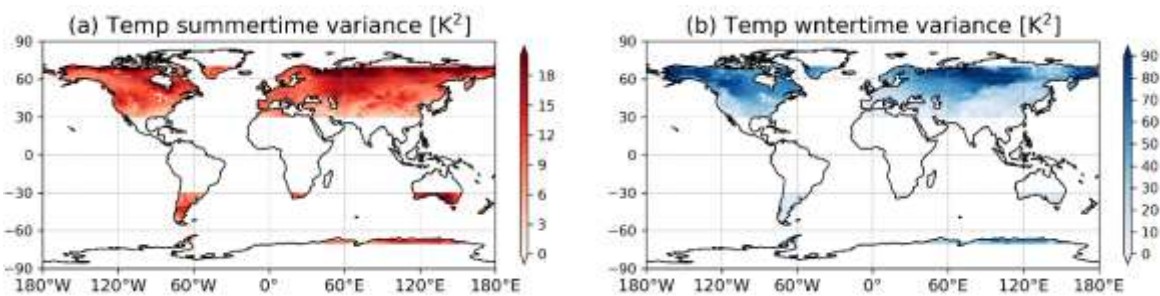

**Figure A1:** Temperature anomaly variance for (a) summer and (b) winter in each hemisphere. The two panels

have differing colour range amplitudes.

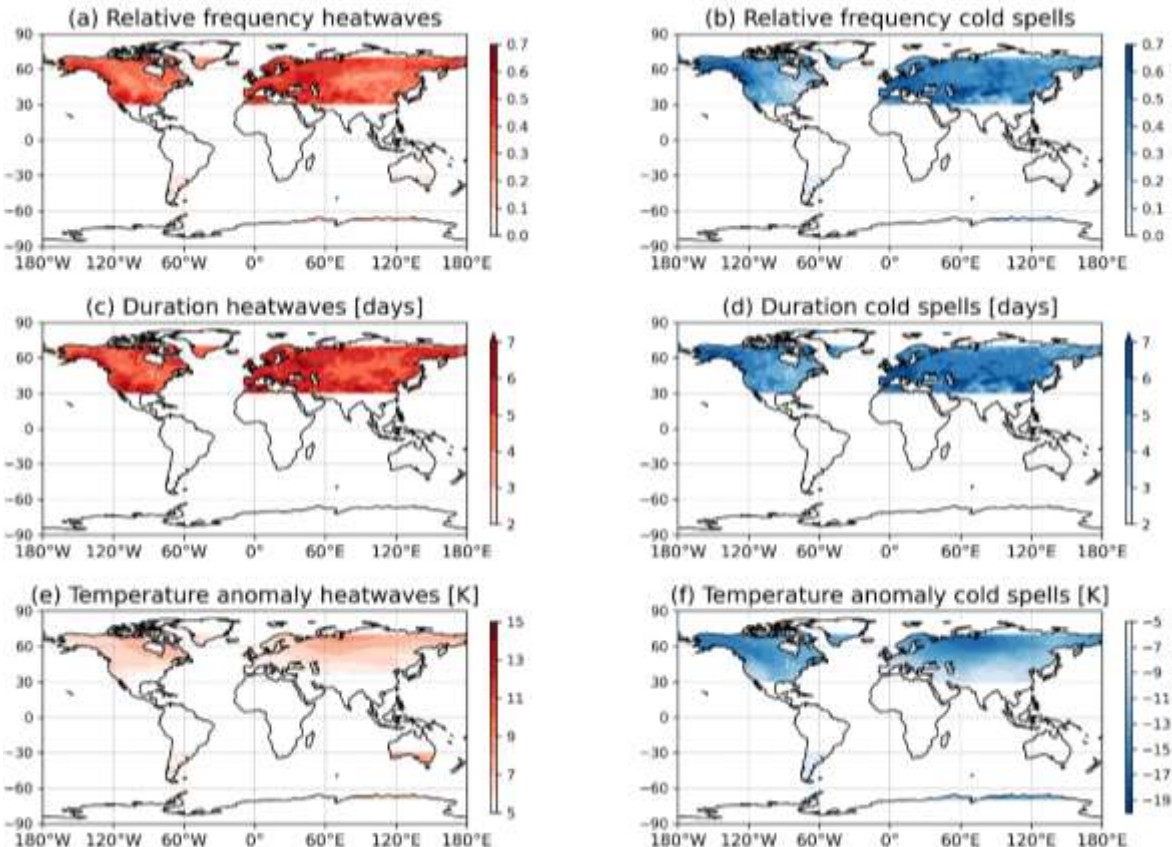

**Figure A2:** As Fig. 2, but for percentile thresholds of 90 (heatwaves) and 10 (cold spells).

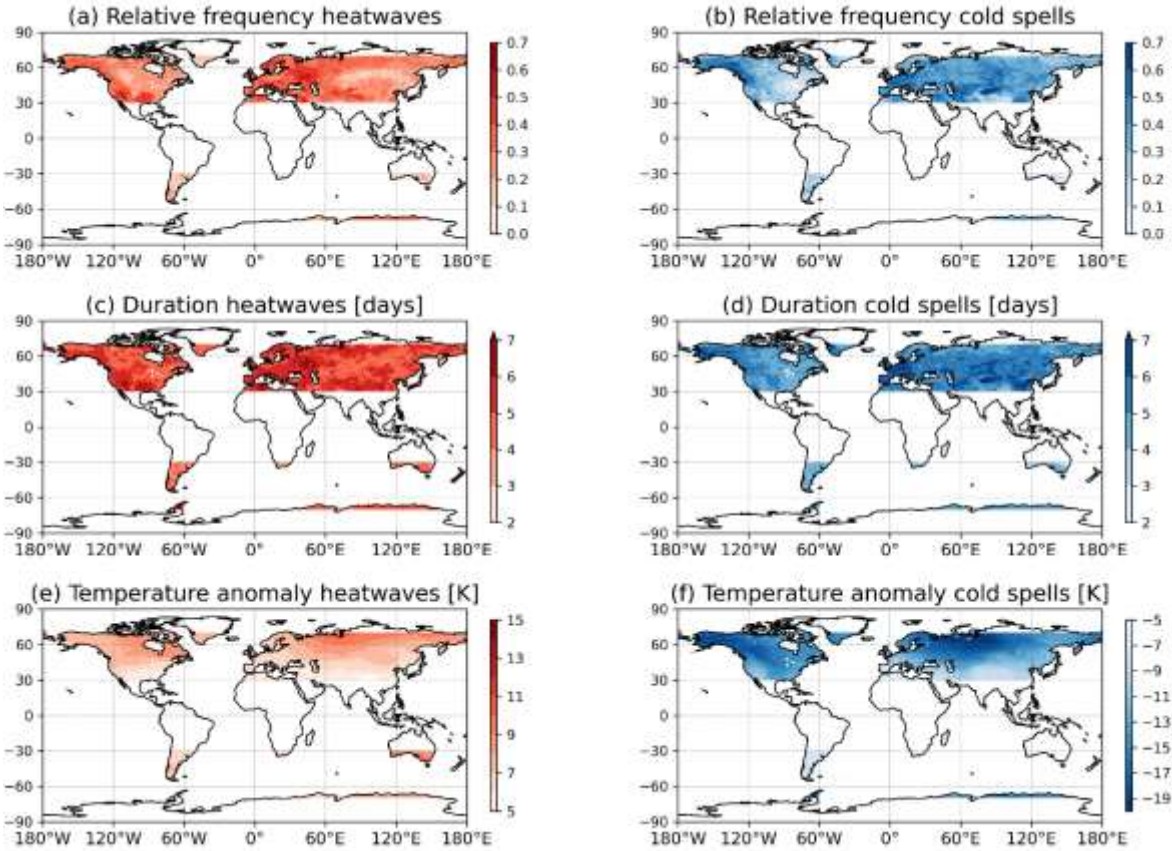

**Figure A3:** As Fig. 2, but for temperature extremes with no requirements for concurrence.

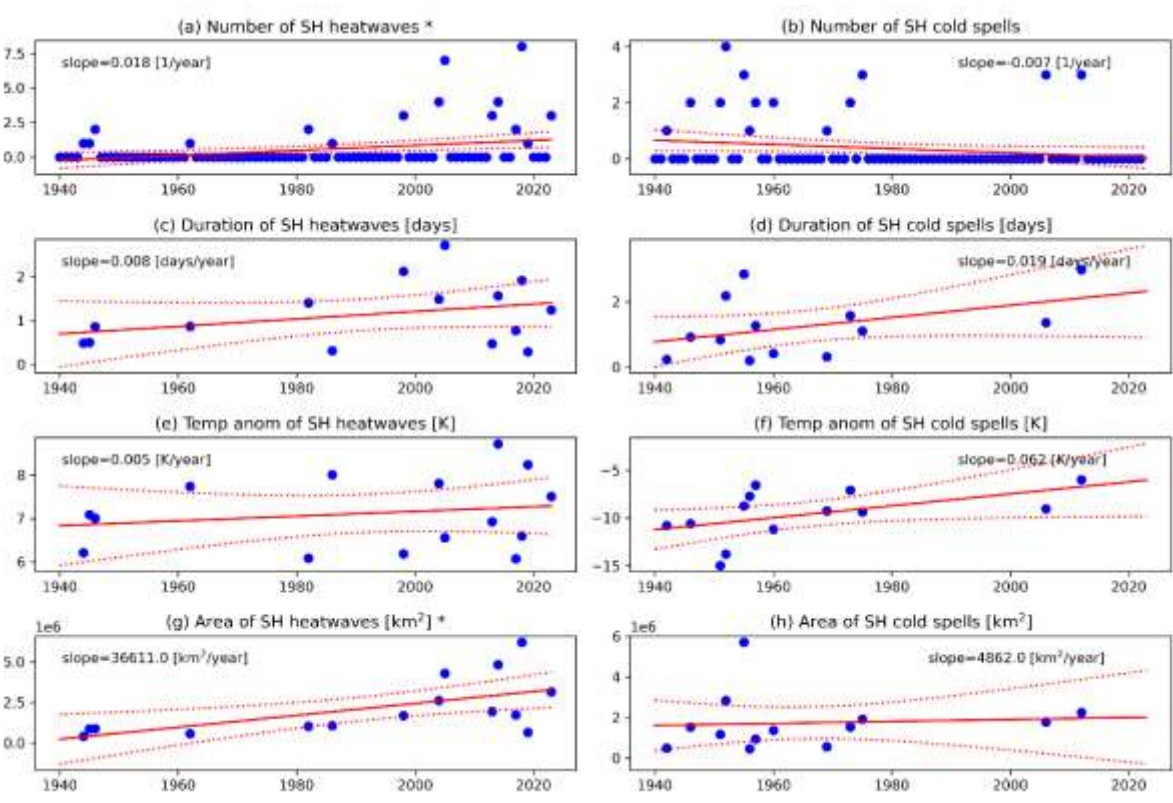

**Figure A4:** As Fig. 3 but for the SH. Seasons without any events are not accounted for when computing the linear fits for duration, severity and extent.

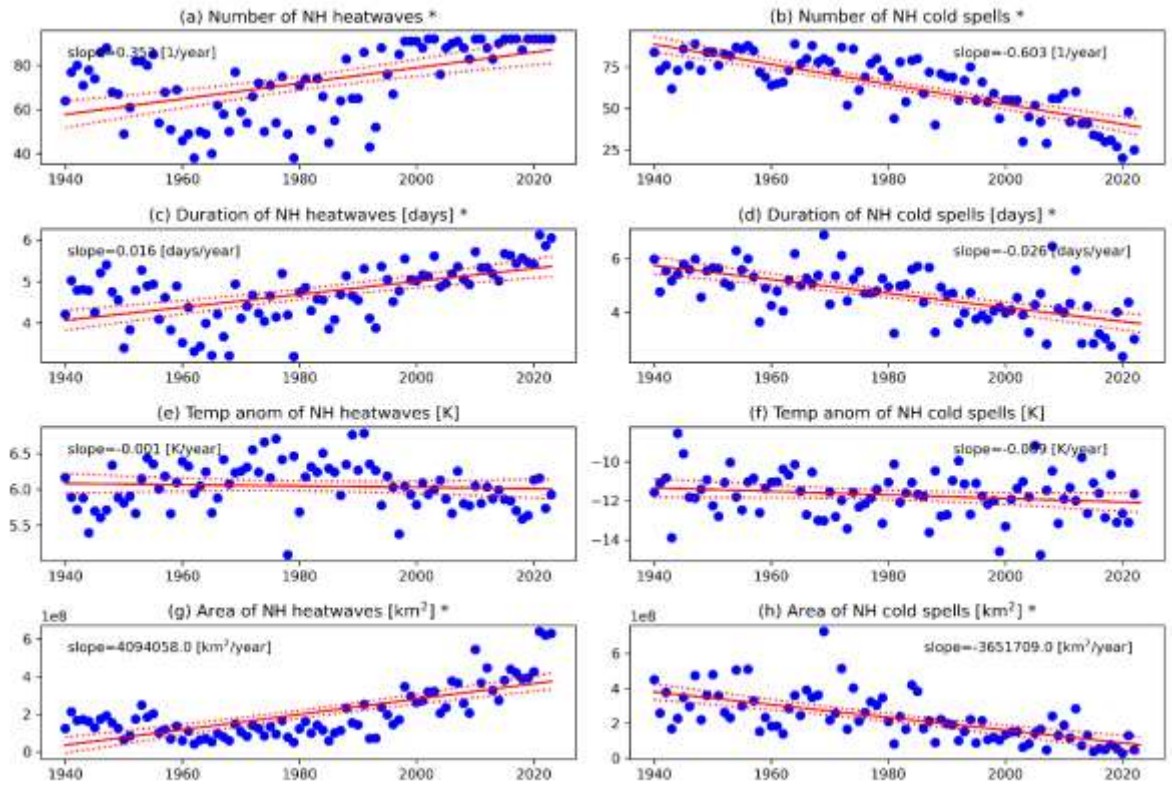

**Figure A5:** As Fig. 3, but for percentile thresholds of 90 (heatwaves) and 10 (cold spells).

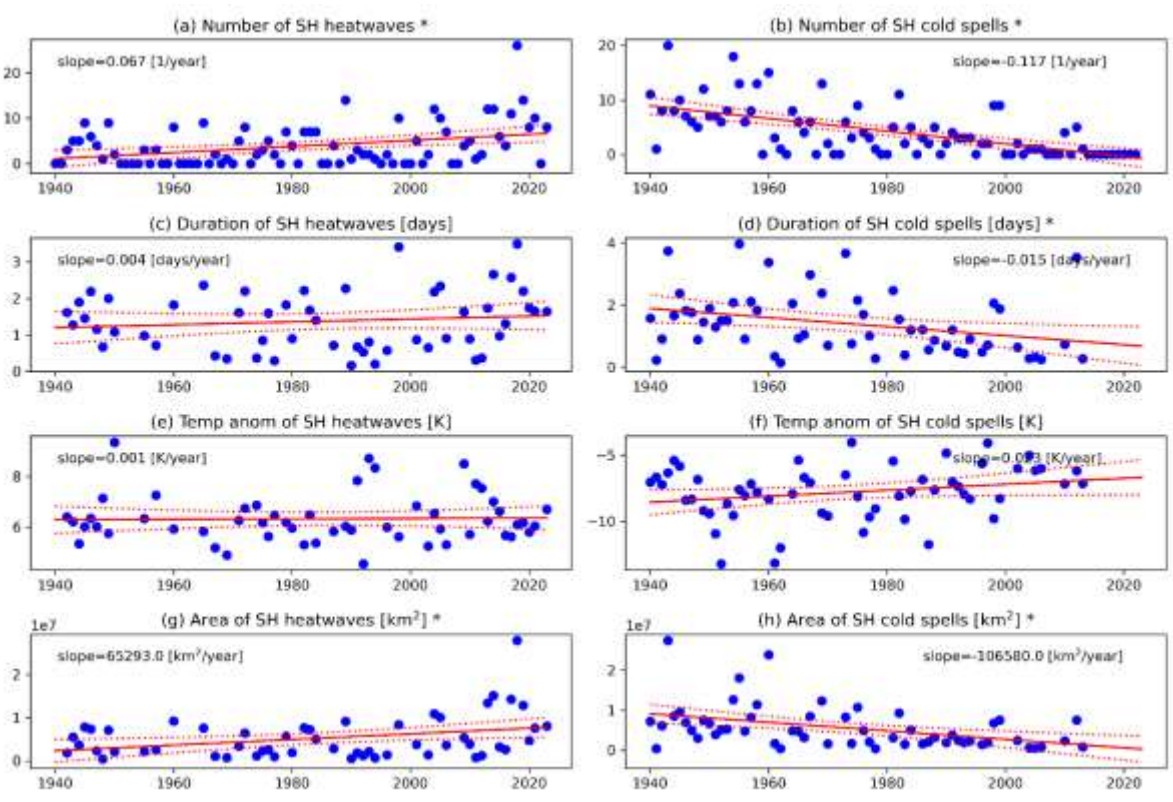

**Figure A6:** As Fig. A4, but for percentile thresholds of 90 (heatwaves) and 10 (cold spells). Seasons without any
events are not accounted for when computing the linear fits for duration, severity and extent.

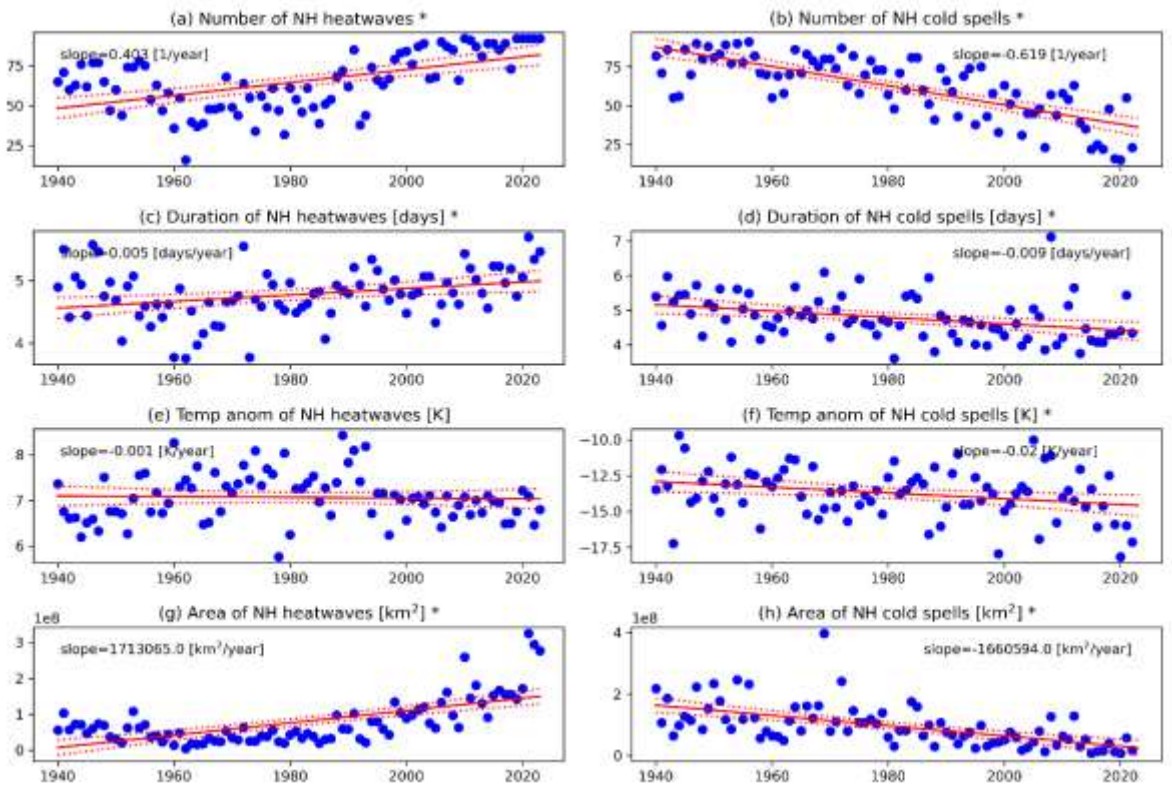

**Figure A7:** As Fig. 3, but for temperature extremes with no requirements for concurrence.

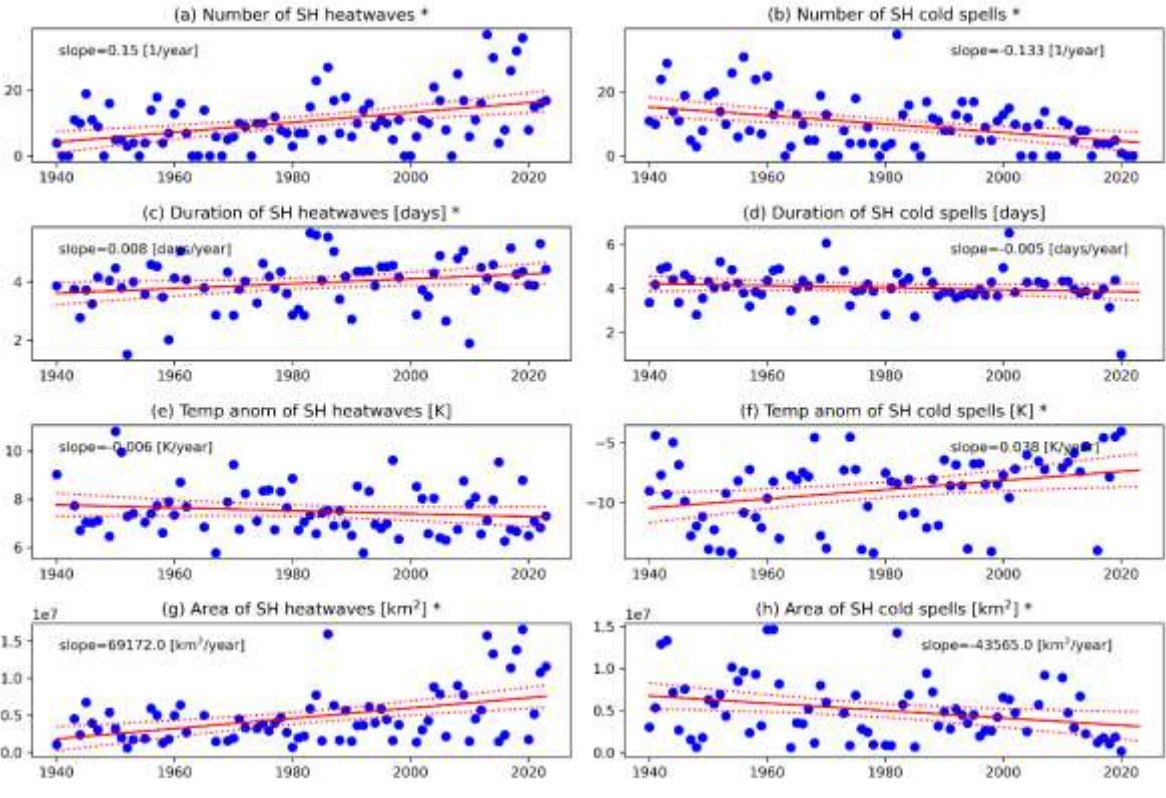

**Figure A8:** As Fig. A4, but for temperature extremes with no requirements for concurrence. Seasons without any events are not accounted for when computing the linear fits for duration, severity and extent.

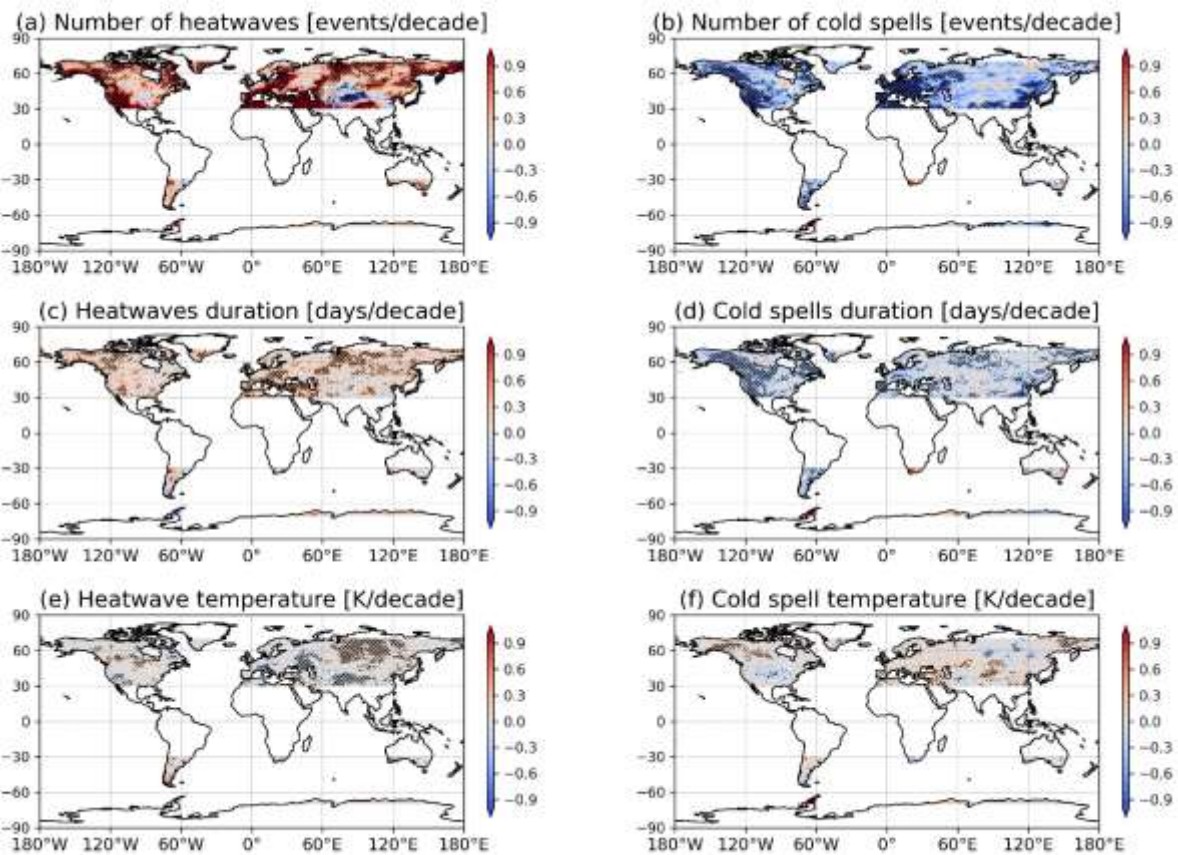

**Figure A9:** As Fig. 4, but for percentile thresholds of 90 (heatwaves) and 10 (cold spells).

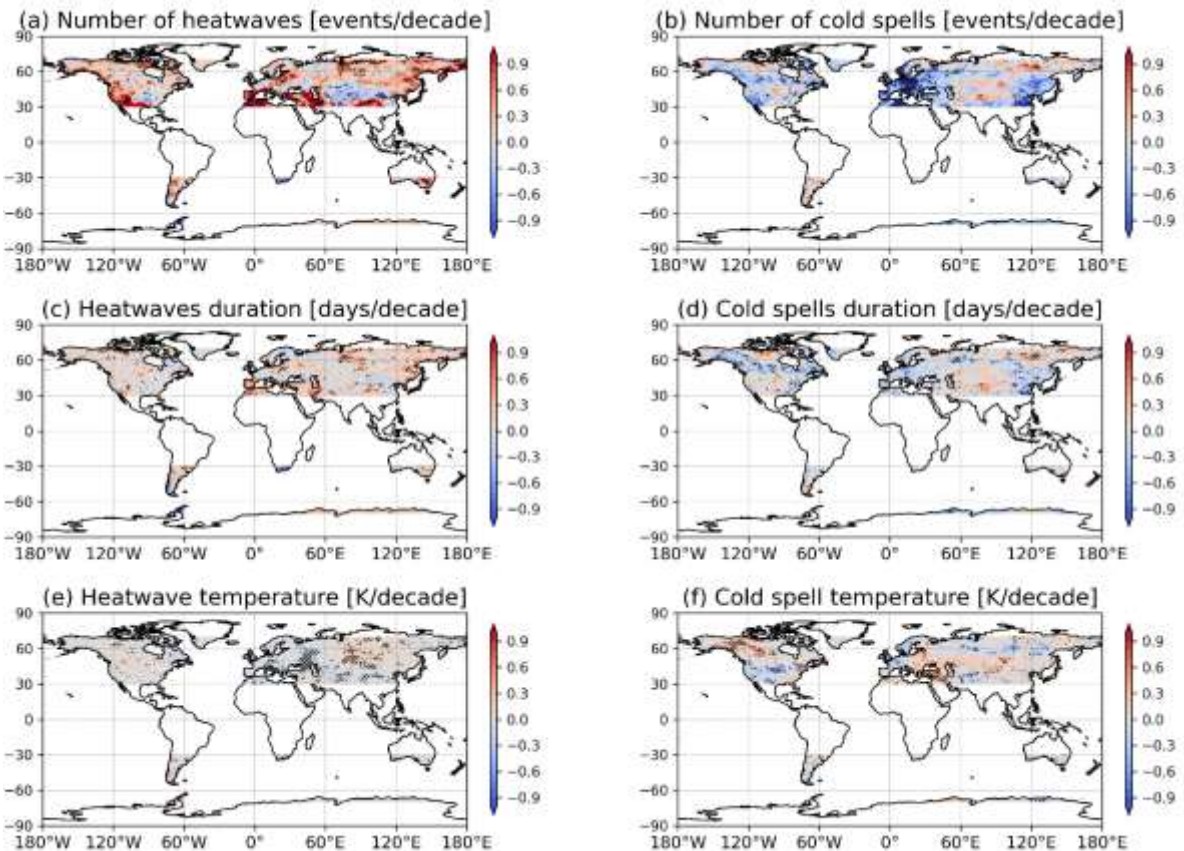

**Figure A10:** As Fig. 4, but for temperature extremes with no requirements for concurrence.

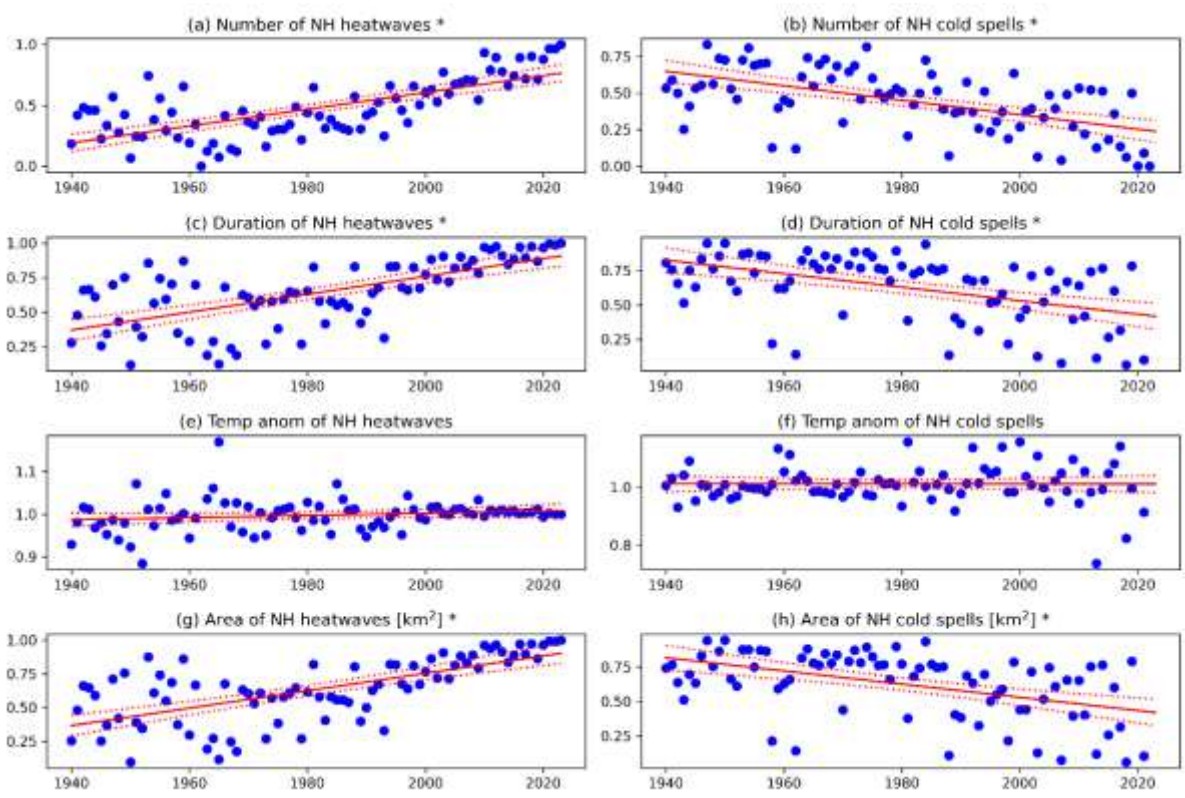

**Figure A11:** As Fig. 3, but for the ratio between concurrent temperature extremes and all temperature extremes.

**Appendix B**

We provide a proof-of-concept for applying our toolbox to variables other than temperature by presenting NH
trend results for 10m daily mean wind extremes in Figs. B1 and B2. These correspond to Figs. 2 and 3 in the main
text, albeit with different values for the toolbox parameters. Compared to compound temperature extremes, the
relative frequency of compound wind extremes is over an order of magnitude lower. Nonetheless, we obtain a
relatively large sample size and find both spatial variations (Fig. B1) and long-term trends (Fig. B2) in the
characteristics of the compound wind extremes.

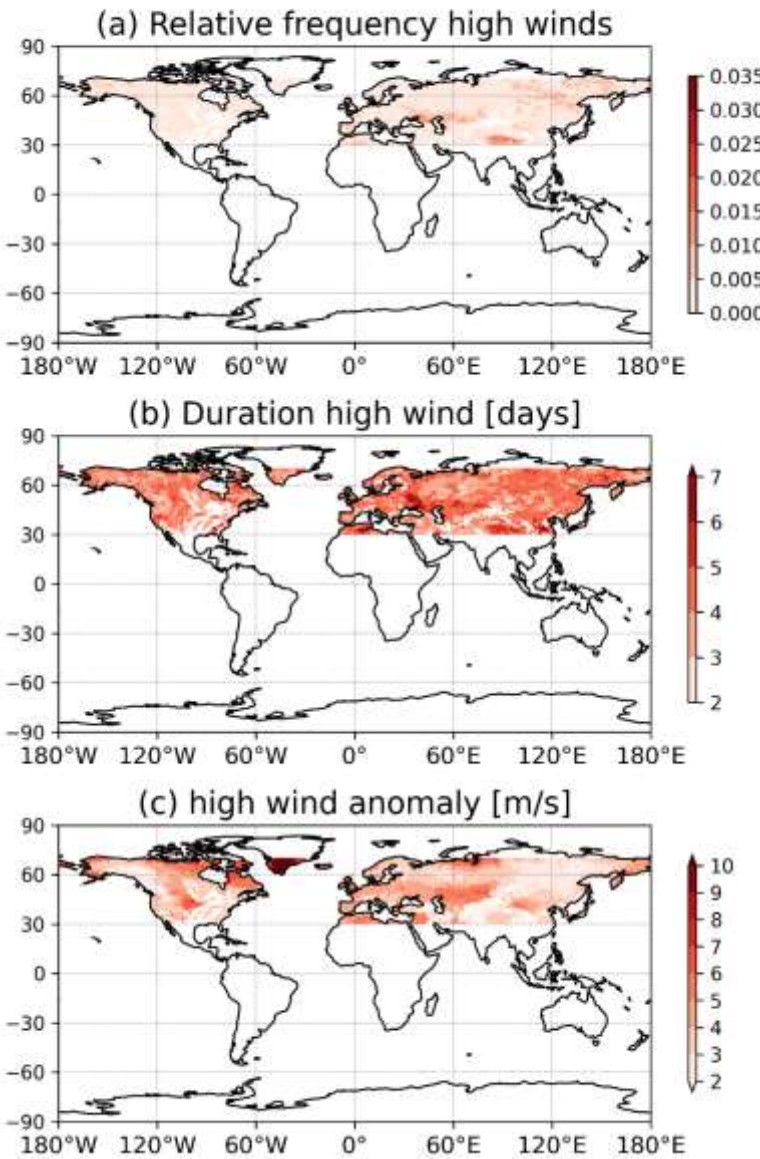

**Figure B1:** As Fig. 2, but for 10m wind. The figure uses the same parameters as for temperature except for a
cluster separation of 250 km and a minimum areal extent of 12,500 km$^2$.

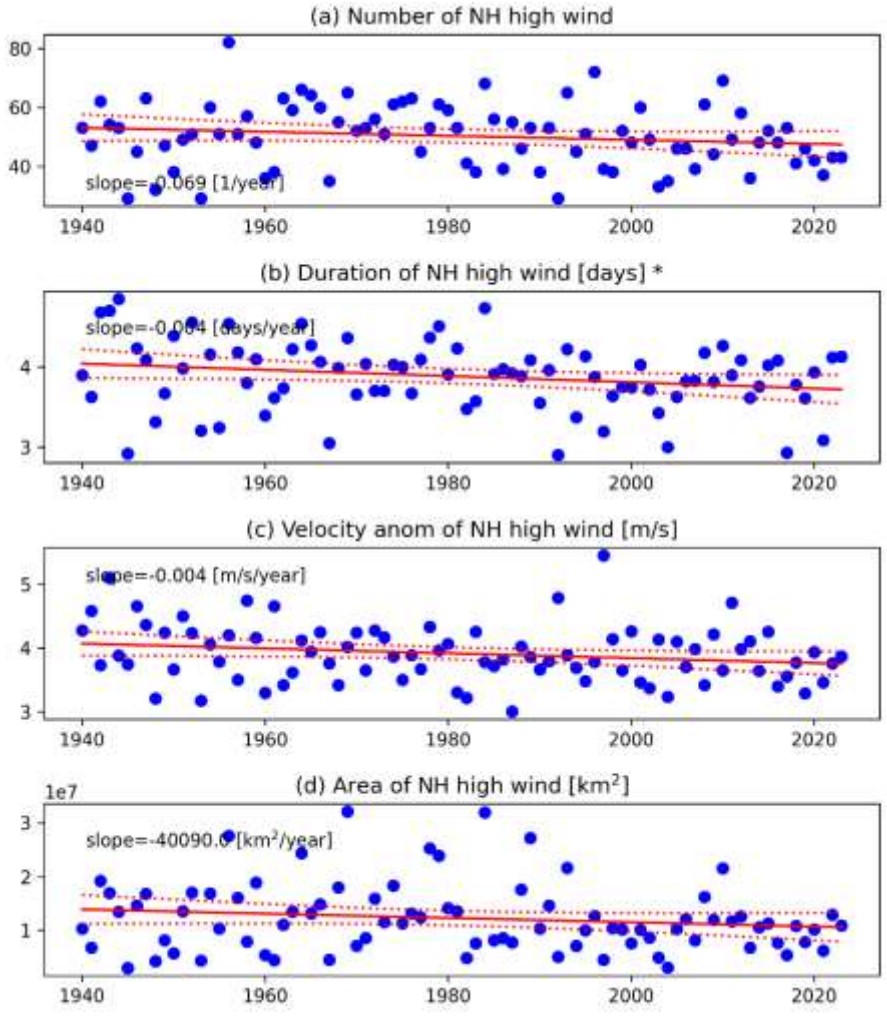

**Figure B2:** As Fig. 3, but for 10m wind. The figure uses the same parameters as for temperature except for a cluster separation of 250 km and a minimum areal extent of 12,500 km$^2$.

**Appendix C**

We provide a parameter sweep of the toolbox (for the parameter pairs minimum cluster separation – minimum duration and minimum areal extent – percentile for temperature extremes). The ranges included in the sweep are summarised in Table 1. Figs. C1 and C2 show linear trends for NH heatwaves and cold spells, respectively, for the different parameter combinations. These correspond to those shown in Fig. 3 in the main text for the reference parameter set (Table 1). The identified trends for heatwaves are generally positive and coherent across the

parameter space (Fig. C1). An exception are the frequency and duration trends for a combination of very large centroid separations and very short minimum durations. In the latter case, we hypothesise that as the global mean temperature increases, it happens more and more often that all heatwaves on a given day are clustered into a single event, and that there are thus fewer and shorter-lasting concurrent heatwaves. We also see negative slopes in heatwave temperature anomaly for combinations of very high percentiles and very large minimum extent values.

We hypothesise that this is due to the fact that, with global warming, such very extreme (both large and intense) heatwaves are increasingly common, and thus their average temperature anomaly decreases.

Cold spells display predominantly negative trends (Fig. C2). A notable exception are positive trends in duration for both very persistent and very intense events, and positive trends in temperature anomaly for large centroid separations. For temperature anomalies, a negative trend implies more intense cold spells, while a positive trend corresponds to a weakening of the cold spells. Imposing large centroid separations presumably eliminates from the sample of concurrent cold spells potentially very intense cold spells limited to a specific sector of the hemisphere, and only retains potentially weaker but truly circum-hemispheric episodes. The positive trend in duration for exceptionally persistent or intense cold spells can be linked to the findings of La Sorte et al. (2021). The authors studied trends in maximum duration of temperature extremes since 1950, and found no systematic decrease for cold spells with global warming, but rather regionally-dependent trends.

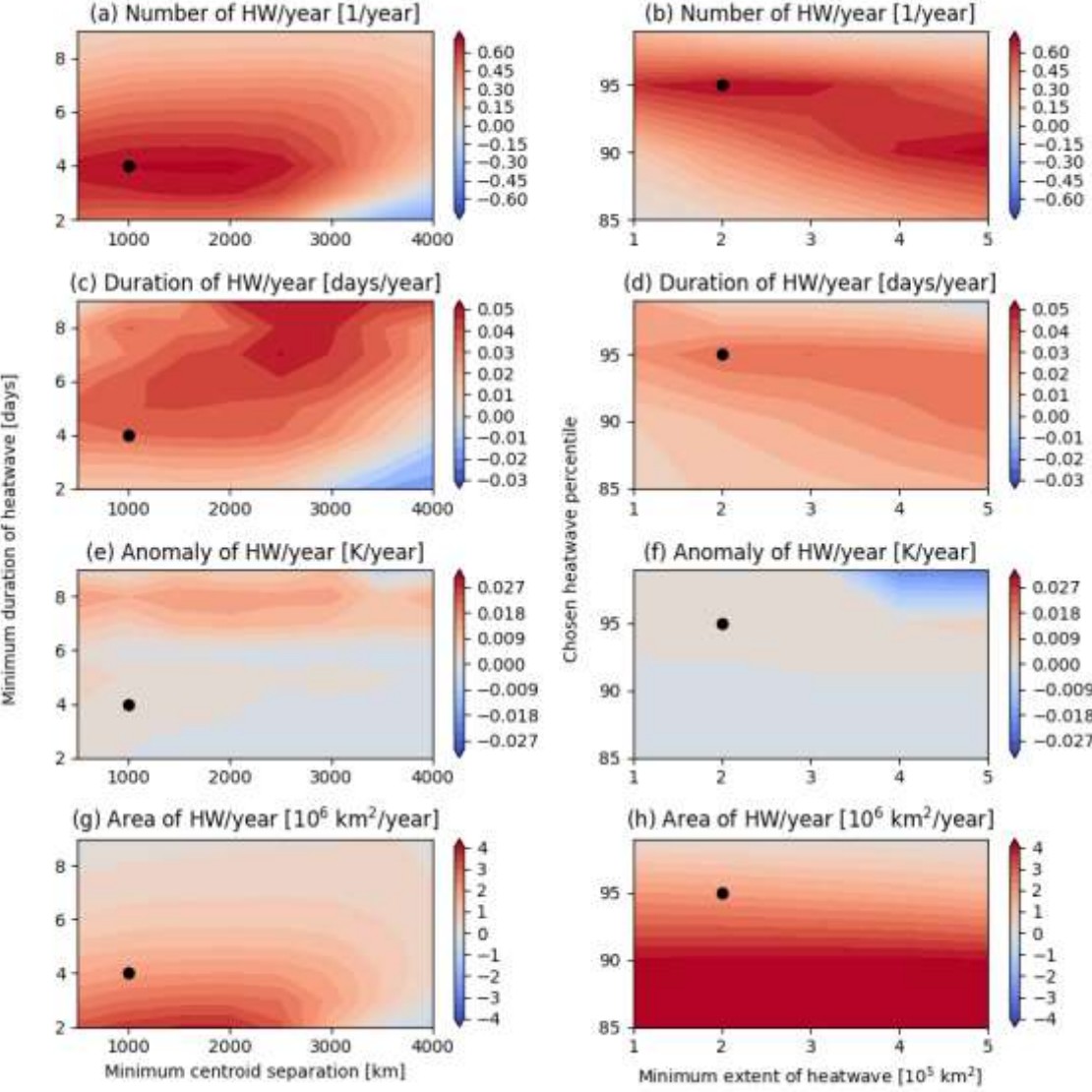

**Fig. C1:** Parameter sweep of the toolbox for the parameter pairs (a, c, e, g) minimum cluster separation – minimum duration and (b, d, f, h) minimum areal extent – percentile for temperature extremes, for concurrent heatwaves. The colours show the linear trends of the following quantities: (a, b) number of heatwaves per year; (c, d) duration of heatwaves (days); (e, f) temperature anomaly of heatwaves (K); and (g, h) heatwave area ($1\times10^5$ km$^2$). The black dots show the parameter combinations used in the paper. If the data for a given parameter combination spans less than 40 years, the trend value is greyed out in the figure.

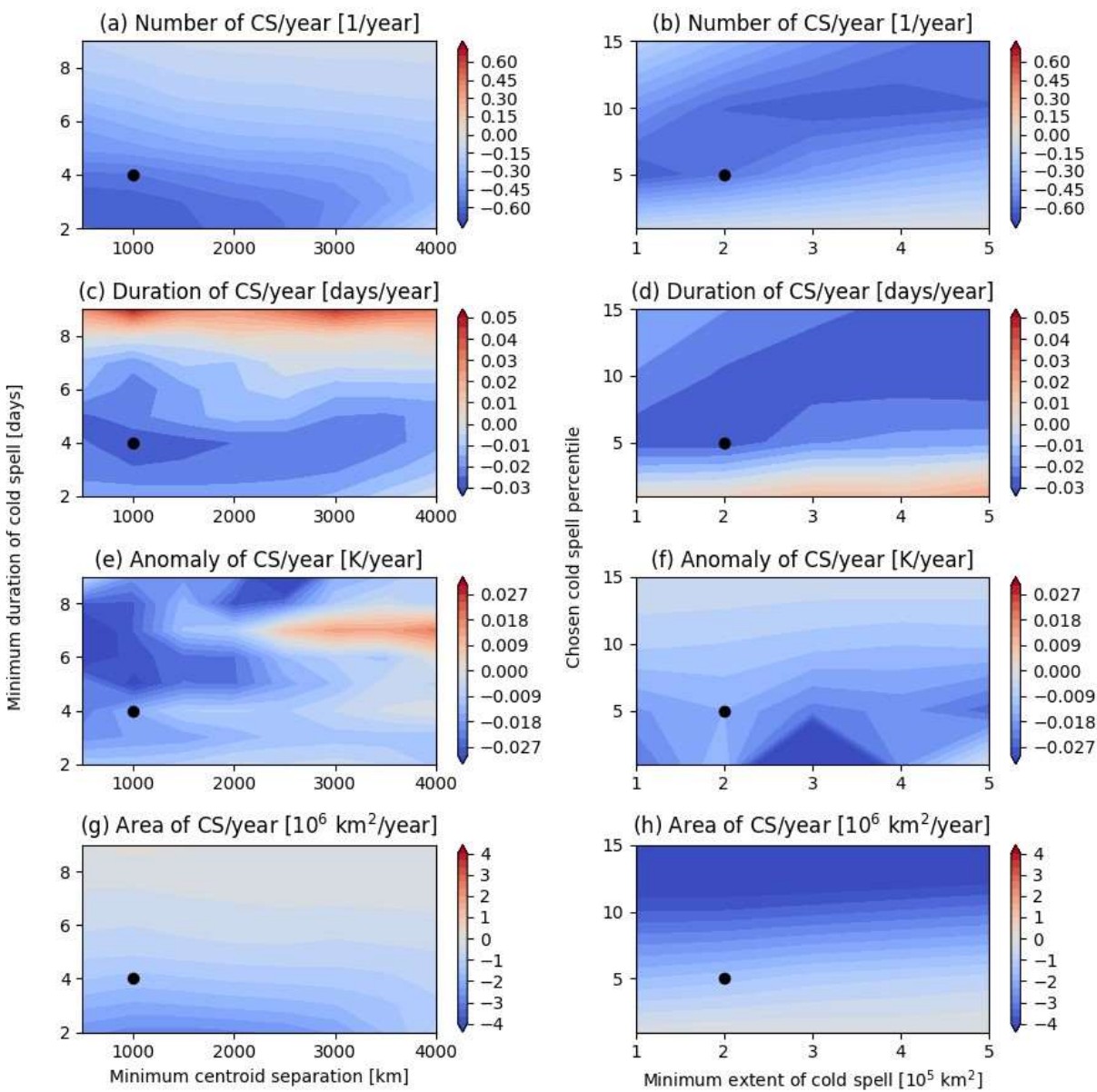

**Fig. C2:** As Fig. C1 but for cold spells.

**Code availability.** The toolbox code is freely available through GitHub: https://github.com/AntSegalini/Global-extreme-events-tool, doi: 10.5281/zenodo.12746817.

**Data availability.** The ERA5 data used in this study is freely available from the Copernicus Climate Change Services Climate Data Store. A sample data file for use with the toolbox is available through GitHub: https://github.com/AntSegalini/Global-extreme-events-tool

**Author contributions.** GM: conceptualization, methodology, formal analysis, visualization, funding acquisition, writing – original draft, writing – review and editing. AS: software, formal analysis, writing – review and editing. AR: conceptualization, writing – review and editing. All authors contributed to the discussion of the results.

**Competing interests.** At least one of the (co-)authors is a member of the editorial board of Earth System Dynamics. The peer-review process was guided by an independent editor, and the authors have no other competing interests to declare.

**Acknowledgements.** This research was supported by the European Union's H2020 research and innovation programme under European Research Council grants no. 948309 and 101112727. AMR was supported by the Helmholtz "Changing Earth" program.

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
