# Peer review of "Climatology and Trends in Concurrent Temperature Extremes in the Global Extratropics"

_Earth System Dynamics, 2023_

## Author Comment (AC4)

(a) Number of HW/year [1/year]

(b) Number of HW/year [1/year]

(c) Duration of HW/year [days/year]

(d) Duration of HW/year [days/year]

(e) Anomaly of HW/year [K/year]

(f) Anomaly of HW/year [K/year]

(g) Area of HW/year [$10^6$ km$^2$/year]

(h) Area of HW/year [$10^6$ km$^2$/year]

Minimum duration of heatwave [days]

Chosen heatwave percentile

Minimum centroid separation [km]

Minimum extent of heatwave [$10^5$ km$^2$]

(a) Relative frequency high winds

(c) Duration high wind [days]

(e) high wind anomaly [m/s]

(a) Number of NH high wind

slope=-0.032 [1/year]

(c) Duration of NH high wind [days]

slope=0.007 [days/year]

(e) Velocity anom of NH high wind [m/s] *

slope=0.021 [K/year]

(g) Area of NH high wind [km²]

slope=888019.0 [km²/year]

(a) Relative frequency heatwaves

(b) Relative frequency cold spells

(c) Duration heatwaves [days]

(d) Duration cold spells [days]

(e) Temperature anomaly heatwaves [K]

(f) Temperature anomaly cold spells [K]

---

## Author Response (AR1)

We provide below updated replies to the Reviewer comments. These largely match those that we provided during the paper discussion phase. However, we have updated some passages to reflect the exact changes that we have implemented in the main paper.

**Reviewer #1**

We thank the Reviewer for their positive outlook on our submission and constructive input. We provide below our replies to the specific comments and the edits that we plan to implement in the manuscript as a result of these.

1. *This paper delves into the concurrent surface temperature extremes using a novel toolbox tailored for understanding such occurrences. The toolbox is well introduced, with clear delineation of the necessary parameters and steps. The authors identify regions with concurrent temperature extremes, characterized by relatively higher frequencies, durations, or temperature anomalies of concurrent extremes. They meticulously analyze the trend of concurrent heatwaves and cold spells across both hemispheres in summer and winter, concluding that the trend of concurrent extremes surpasses that of individual temperature extremes. Given the characterization of this manuscript, the presentation of temperature extreme trends is comprehensive, logically structured, and well-organized.*

Thank you for your positive feedback, we are glad that you found the manuscript clear and well-structured.

2. *However, one aspect warrants attention: the focus of the letter. It's imperative to clarify whether the primary emphasis lies in analyzing trends of concurrent temperature extremes or in introducing and propagating the toolbox. The statement in lines 107-108 regarding not attempting a full parameter sweep but rather selecting values for comparison with previous literature suggests an intent to showcase the toolbox's effectiveness in reproducing prior results. If so, revisiting the title and abstract might be beneficial. Alternatively, if the letter aims to highlight trends in concurrent temperature extremes, it would be advisable to incorporate parameter sensitivity tests and potentially explore additional content, such as dynamical feature trends in the upper-troposphere (if feasible within the toolbox). Such enhancements would underscore the toolbox's efficacy for this type of research.*

The Reviewer raises an important point. Our aim with this manuscript is to present a tool that other researchers in the field of compound extremes may use for their own research. In this perspective, our priority is to introduce and disseminate the toolbox. Nonetheless, we believe that to achieve this we need to showcase the type of analyses that may be conducted with the toolbox, hence the importance of including a relatively detailed description of the climatology of compound temperature extremes and their trends. To address the concern of the Reviewer, we have added figures displaying a parameter sweep of the toolbox (new Appendix C). We specifically vary the parameters (cluster separation and duration) and (minimum cluster area and percentile), and summarise the results at hemispheric scale for the different parameter combinations. We copy the figures for heatwaves and cold spells below. These are included

in the new Appendix C in the paper, and referenced in the main text. The identified trends for heatwaves are generally positive and coherent across the parameter space (Fig. C1). An exception are the frequency and duration trends for a combination of very large centroid separations and very short minimum durations. In the latter case, we hypothesise that as the global mean temperature increases, it happens more and more often that all heatwaves on a given day are clustered into a single event, and that there are thus fewer and shorter-lasting concurrent heatwaves. We also see negative slopes in heatwave temperature anomaly for combinations of very high percentiles and very large minimum extent values. We hypothesise that this is due to the fact that, with global warming, such very extreme (both large and intense) heatwaves are increasingly common, and thus their average temperature anomaly decreases. Cold spells display predominantly negative trends (Fig. C2). A notable exception are positive trends in duration for both very persistent and very intense events, and positive trends in temperature anomaly for large centroid separations. A positive trend in temperature anomaly corresponds to a weakening of the cold spells. Imposing large centroid separations presumably eliminates from the sample of concurrent cold spells potentially very intense cold spells limited to a specific sector of the hemisphere, and only retains potentially weaker but truly circum-hemispheric episodes. The positive trend in duration for exceptionally persistent or intense cold spells can be linked to the findings of La Sorte et al. (2021). The authors studied trends in maximum duration of temperature extremes since 1950, and found no systematic decrease for cold spells with global warming, but rather regionally-dependent trends.

Concerning the final part of the Reviewer's comment, the toolbox is unfortunately not suited to the detection of dynamical features in the upper troposphere (i.e. the drivers of the extremes), but only tailored to the detection of the extremes themselves. We will specify this at the beginning of Sect. 4 in the revised manuscript.

[Figure]

**Fig. C1:** Parameter sweep of the toolbox for the parameter pairs (a, c, e, g) minimum cluster separation – minimum duration and (b, d, f, h) minimum areal extent – percentile for temperature extremes, for concurrent heatwaves. The colours show the linear trends of the following quantities: (a, b) number of heatwaves per year; (c, d) duration of heatwaves (days); (e, f) temperature anomaly of heatwaves (K); and (g, h) heatwave area (1×$10^5$ km$^2$). The black dots show the parameter combinations used in the paper.

[Figure]

**Fig. C2:** As Fig. C1 but for cold spells.

**Reviewer #2**

We thank the Reviewer for their positive outlook on our submission and constructive input. We provide below our replies to the specific comments and the edits that we plan to implement in the manuscript as a result of these.

**General Comments**

1. *My biggest concern regarding the present state of the manuscript is an inadequately explained definition of concurrent HW (CS) and its potential conflict with toolbox's clustering algorithm. If I understand correctly from the Figure 2 caption, a HW (CS) is regarded as concurrent if another HW (CS) is present within the selected domain (northern/southern mid-latitudes). It seems that a resulting frequency of concurrent HW (CS) would differ considerably under different clustering and minimum extend settings (Figure 1c). For example, if the clustering distance and minimum area are small, almost every HW (CS) would be accompanied by other events, resulting in too many concurrent extremes. By contrast, if the clustering distance and minimum area are set as large, the algorithm would merge HW (CS) thorough the domain, resulting in too little concurrent extremes. In my opinion, the usage of the toolbox would require quite a lot of expert judging to capture concurrent extremes properly. Before publishing the toolbox, authors should consider adding default parameters or, alternatively, their recommended range for individual extreme events.*

The Reviewer is absolutely correct in stating that the frequency of concurrent HW (CS) would differ considerably under different clustering and minimum extent settings. Indeed, there is no single objective definition of temperature extremes, and the most relevant parameters will depend on the user case being analysed. For example, a researcher working on health impacts of extreme temperatures may be more interested in compound heatwaves or cold spells that are small in extent and close to each other (since healthcare systems predominantly work on a national basis), while a researcher looking at global food supply may be interested in larger and well-distanced heatwaves or cold spells (since the impacts of temperature extremes in different continents may in this case compound). We believe that there is no such thing as too many or too few concurrent extremes in absolute terms, but rather that what "too many" or "too few" means depends heavily on who will use our toolbox. We have clarified this viewpoint at the beginning of Sect. 4 of the revised manuscript.

We nonetheless agree with the Reviewer that it is important for users to have an idea of how sensitive the results are to the choice of toolbox parameters. To address this point, we have added two figures displaying a parameter sweep of the toolbox (new Appendix C). We specifically vary the parameters: (cluster separation and duration) and (minimum cluster area and percentile), and summarise the results at hemispheric scale for the different parameter combinations. We copy the figures for heatwaves and cold spells below. These are included in the new Appendix C in the paper, and referenced in the main text. The identified trends for heatwaves are generally positive and coherent across the parameter space (Fig. C1). An exception are the frequency and duration trends for a combination of very large centroid separations and very short minimum durations. In the latter case, we hypothesise that as the global mean temperature increases, it happens more and more often that all heatwaves on a given day are clustered into a single event, and that there are thus fewer and shorter-lasting concurrent heatwaves. We also see negative slopes in heatwave temperature anomaly for combinations of very high percentiles and very large minimum extent values. We hypothesise that this is due to the fact that, with global warming, such very extreme (both large and intense) heatwaves are increasingly common, and thus their average temperature anomaly decreases. Cold spells display predominantly negative trends (Fig. C2). A notable exception are positive trends in duration for both very persistent and very intense events, and positive trends in temperature anomaly for large centroid separations. A positive trend in temperature anomaly corresponds to a weakening of the cold spells. Imposing large centroid separations

presumably eliminates from the sample of concurrent cold spells potentially very intense cold spells limited to a specific sector of the hemisphere, and only retains potentially weaker but truly circum-hemispheric episodes. The positive trend in duration for exceptionally persistent cold spells. The positive trend in duration for exceptionally persistent or intense cold spells can be linked to the findings of La Sorte et al. (2021). The authors studied trends in maximum duration of temperature extremes since 1950, and found no systematic decrease for cold spells with global warming, but rather regionally-dependent trends.

Concerning the final part of the Reviewer's comment, the toolbox is unfortunately not suited to the detection of dynamical features in the upper troposphere (i.e. the drivers of the extremes), but only tailored to the detection of the extremes themselves. We will specify this at the beginning of Sect. 4 in the revised manuscript.

[Figure]

**Fig. C1:** Parameter sweep of the toolbox for the parameter pairs (a, c, e, g) minimum cluster separation – minimum duration and (b, d, f, h) minimum areal extent – percentile for temperature extremes, for concurrent heatwaves. The colours show the linear trends of the following quantities: (a, b) number of heatwaves per year; (c, d) duration of heatwaves (days);

(e, f) temperature anomaly of heatwaves (K); and (g, h) heatwave area (1×10$^5$ km$^2$). The black dots show the parameter combinations used in the paper.

[Figure]

**Fig. C2:** As Fig. C1 but for cold spells.

Finally, we added a table summarising the default parameters that we used for the results displayed in the main text and for the parameter sweep shown in Appendix C (new Table 1). The values used for the results displayed in the main text were identified as commonly-used or middle-of-the-range parameters from the past literature on temperature extremes.

*2. Additionally, authors may consider evaluating other variables then temperature (Line 50), to enhance interdisciplinarity which is required by the ESD journal.*

This is a good idea. We have now tested the toolbox on 10m wind data, and show the results below. We have added these figures in the new Appendix B, and reference the Appendix in the main text. We chose not include a figure corresponding to Fig. 4 in the paper because 10m wind extremes are much smaller in scale than temperature extremes and the figure was spatially noisy and not very informative.

Compared to compound temperature extremes, the relative frequency of compound wind extremes is over an order of magnitude lower. Nonetheless, we obtain a relatively large sample size and find both spatial variations (Fig. B1) and long-term trends (Fig. B2) in the characteristics of the compound wind extremes.

[Figure]

**Figure B1:** As Fig. 2, but for 10m wind. The figure uses the same parameters as for temperature except for a cluster separation of 250 km and a minimum areal extent of 12,500 km².

[Figure]

**Figure B2:** As Fig. 3, but for 10m wind. The figure uses the same parameters as for temperature except for a cluster separation of 250 km and a minimum areal extent of 12,500 km².

**Regarding the toolbox**

1. The authors plan to publish the toolbox's python code after acceptance (Line 271). I am not an expert in programming, so I am wondering if it is possible to add a more user-friendly graphical interface to make the toolbox available for a broader community. Please comment.

It is true that adding a Graphical User Interface (GUI) facilitates the first approach to the toolbox. However, it also forces the user to do only a limited set of operations that a programmer has defined a priori. The toolbox is at the moment a library of python functions that can be used sequentially (or not) where significant flexibility is granted to the user by the

script implementation of Python. As we mentioned in our replies in the discussion phase, we do not have extensive experience in developing GUIs. One of the authors, A. Segalini, made some initial attempts at this, but we concluded that developing a high-quality GUI is a major undertaking. What we would realistically have the time for is a basic GUI with limited functionalities, which we think would detract from out toolbox by giving a misleading impression of the real potentialities of the toolbox.

*2. The toolbox is planned to be published via authors' GitHub (Line 272). Please make sure that it will receive a permanent DOI identifier.*

GitHub now allows to generate DOIs through Zenodo, and we will ensure that our code is published and a DOI is assigned to the code version as used in the final iteration of this manuscript prior to final acceptance. We have in the meanwhile made the code publicly available through GitHub without a DOI: [https://github.com/AntSegalini/Global-extreme-events-tool](https://github.com/AntSegalini/Global-extreme-events-tool)

**Regarding the climatological analysis**

*1. Line 112: 'roughly 40% of single-gridbox heatwaves at those locations are part of a set of multiple concurrent, large-scale heatwaves across the NH' – if I understand the definition correctly, only another single heat wave (related or not) is sufficient to fulfil the criterion for a concurrent event. Therefore, in my opinion, 'across the NH' is exaggerated.*

The Reviewer has understood the definition correctly and we agree that "across" is not the right term to use here. We have updated this to "in the NH".

*2. Line 130–132: The explanation regarding a duration of compound (concurrent?) extreme is hard to follow. Please rewrite for better clarity.*

We have rephrased this passage as follows: "Note that, while we impose a minimum duration threshold for single-gridbox extremes, the duration of a compound temperature extreme can be shorter than this. Indeed, for compound extremes we only count the duration while there are two or more concurrent extremes – even if this temporal overlap period is shorter than the minimum duration threshold for single-gridbox extremes."

*3. Line 153: 'Concurrent cold spells show mirror trends' – Not everywhere. For example, negative HW and CS trends (blue colour) can be found in parts of the US, while positive HW and CS trends are in parts of Siberia and Scandinavia. Please modify the wording.*

Thank you for spotting this. We have rephrased this passage as follows: "Concurrent cold spells mostly mirror these trends, with significant decreases in number, duration and area (Fig. 3b, d, h). However, there are regions where both extremes show similar trends, for example in parts of Central Asia and Siberia, central North America and Scandinavia".

*4. Figure 4: I suppose these are linear trends (?) Please clarify in the caption.*

The Reviewer is correct and we have updated the caption accordingly.

> 5. *Line 168: It should be mentioned that the large HW trend in Western North America is probably related mainly to the extraordinary 2021 heat wave*

This is an interesting point, as a particularly extreme heatwave could indeed affect the regional trends, but we do not think this is the case for Western North America. Indeed, while the 2021 heatwave was unprecedented in terms of magnitude of the temperatures, it was not exceptionally long-lived. The extreme part of the heatwave lasted for roughly a week (see e.g. Pons et al., 2024). Thus, if this heatwave were to explain the large HW trend in Western North America, we would expect to see the largest trends in the HW temperatures. However, as shown in Fig. 2, the Western North American hotspot is mainly in the frequency and duration of heatwaves, and is less evident when it comes to magnitude. To verify this hypothesis, we have reproduced Fig. 2 in the manuscript but excluding 2021 from the analysis. We show this in Fig. R1 below. We argue that the differences compared to the figure shown in the paper, which includes 2021, are minor and that the 2021 heatwave thus does not explain the regional trends seen in Western North America.

[Figure]

**Fig. R1:** As Fig. 2 in the manuscript, but excluding 2021 from the analysis.

Pons, F. M. E., Yiou, P., Jézéquel, A., & Messori, G. (2024). Simulating the Western North America heatwave of 2021 with analog importance sampling. *Weather and Climate Extremes*, 100651.

6. *I believe that the southern domain should be cropped to exclude Antarctica's coast and its shelf ice*

We understand the Reviewer's doubts about including Antarctica in our analysis. However, we note two points to justify our choice: (i) We apply a land mask as part of our toolbox, so shelf ice is automatically excluded from our analysis of temperature extremes. (ii) Antarctica has experienced very large temperature extremes in recent years, including what has been termed "the largest ever-recorded heatwave" in 2022 which received the attention of the scientific community (e.g. Blanchard-Wrigglesworth et al., 2023, Wille et al., 2024), so we believe that including parts of Antarctica is relevant for our climatological analysis. If this were not the case for specific users (or if a specific user wanted to include the whole of Antarctica), the toolbox offers the flexibility to adjust the latitudinal domain as desired.

Blanchard-Wrigglesworth, E., Cox, T., Espinosa, Z. I., & Donohoe, A. (2023). The largest ever recorded heatwave—Characteristics and attribution of the Antarctic heatwave of March 2022. *Geophysical Research Letters*, *50*(17), e2023GL104910.

Wille, J. D., and Coauthors, (2024) The Extraordinary March 2022 East Antarctica "Heat" Wave. Part I: Observations and Meteorological Drivers. J. Climate, 37, 757–778, https://doi.org/10.1175/JCLI-D-23-0175.1.

---

## Author Response (AR2)

**ESD-2023-45 Response to the Editor**

Dear Editor,

We have now implemented the requested technical corrections and have provided a doi for our code in the appropriate section. We have further updated figures B1 and B2 which mistakenly showed results for the 90$^{th}$ percentile instead of the 95$^{th}$ percentile. The differences in the results are minimal, and these figures were anyways provided as examples only and are not discussed in the main text.

Best Regards,

Gabriele Messori on behalf of all co-authors